# Can venture capital shareholding improve M&A performance? An empirical study based on Chinese GEM-listed companies

**Kun Chao, Meijia Wang, Yanyong Hu** *, **Shixue Wang**

School of Management, China University of Mining and Technology (Beijing), Beijing, China

* huyanyong615@163.com

**Data Availability Statement:** All data required are available from the Figshare database (accession DOI: https://doi.org/10.6084/m9.figshare.25902694.v2).

## Abstract

Existing studies have explored the impact of venture capital shareholding on the GEM-listed companies before and after listing from multiple perspectives. However, there has been limited research on the influence of venture capital shareholding on these companies' mergers and acquisitions(M&A) activities and performance. Additionally, two conflicting research findings have been presented in limited relevant studies. In order to clarify the mechanism by which venture capital shareholding affects M&A activities and performance of GEM-listed companies and verify existing research conclusions, this paper takes 468 M&A events completed by the acquirer of China's GEM-listed companies between 2014 and 2016 as samples to explore venture capital shareholding's effects on the M&A performance of GEM-listed enterprises. The empirical findings demonstrate that GEM-listed enterprises with venture capital shareholding perform significantly better in terms of short-term and long-term M&A performance than those without; with the increase in venture capital shareholding ratio, the short-term M&A performance of GEM-listed enterprises has remarkably improved, but the long-term M&A performance does not show obvious correlation; joint investment of venture capital can significantly improve the short-term M&A performance of GEM-listed enterprises, but it has no substantial influence on long-term M&A performance. Based on further analysis of the empirical study, it is concluded that the common one-share ownership structure of GEM-listed enterprises is not conducive to the play of the monitoring function of venture capital, and the insufficient incentives and free-riding thinking also weaken the motivation and input of some venture capital shareholders to provide value-added services. This study systematically elucidates the mechanism and impact of venture capital shareholding on the M&A performance of GEM-listed companies, addressing the shortcomings in existing research. It is conducive for GEM-listed companies to gain a rational understanding and effectively leverage the active role of venture capital shareholders in M&A activities.

**Funding:** This research was funded by the Basic Research Funds of China University of Mining and Technology (Beijing) - Fund for Cultivation of Top notch Innovative Talents for Doctoral Graduates (No. BBJ2023046). The funder is the corresponding author Yanyong Hu of this paper, who plays important roles in the methodology, validation, formal analysis, project administration, data curation, funding acquisition, and writing—review & editing.

**Competing interests:** The authors have declared that no competing interests exist.

## 1. Introduction

In recent years, as the roles of venture capital in promoting entrepreneurship and innovation and boosting the growth of entrepreneurial enterprises have become increasingly apparent [1], research on how it creates value for invested enterprises has also emerged. At present, relevant studies mainly focus on exploring venture capital's functions in pre-listing and the listing process of the invested companies [2], while its role in the post-listing process of the enterprise is less involved. Once the funded company goes public, venture capital has the primary conditions for withdrawal and exit. Still, the setting of the equity lockup period [3] makes it unrealizable in a short-term period. After the lockup period is released, venture investors will continue to hold these shares if there is still potential for growth at a reasonable value. Relevant studies show that a mere 6.8% of the venture capital withdraws and exits after one year of the invested company's listing, only 42% of the venture capital withdraws and exits after three years of the listing [4], and only 44% within five years [5]. It can be seen that after the invested enterprise has gone public, most venture capitalists choose to continue to hold their shares for varying periods and play their roles [6].

Owing to the relatively stable ownership of venture capital, the extent of studies regarding the roles of venture capital on the value creation of the invested enterprises is gradually expanding from the pre-listing and the listing process to the post-listing. Some scholars at home and abroad have researched the impact of venture capital on investing and financing behaviors [5], earnings management [7], establishing an efficient board of directors [8], improving corporate governance structures [9], implementing equity incentives and replacing CEO [10], technological innovation [11–14] after going public and other aspects. However, up to now, scholars have yet to pay attention to its impact on invested companies' post-listing M&A activities and M&A performance.

Good development potential often enables GEM-listed companies to obtain over-raised funds of varying sizes during the IPO process, and the pursuit of corporate growth also provides GEM-listed companies with enough incentives to seek external M&A, which is the most significant operating activity of an enterprise, as well as the easiest to be noticed by investors. As institutional investors with professional management capacity and rich experience in capital market operation, venture capitalists should actively participate in the M&A activities of invested companies, promote the success of M&A, and improve companies' M&A performance by playing roles in certification [15–17], monitoring [18, 19], value-added services [20]. However, due to the minimal research literature in this area, two entirely different findings are presented concerning venture capital's effects on GEM-listed acquirers' M&A performance: One conclusion suggests that M&A performance over the short and long terms for GEM-listed companies with venture capital shareholding become significantly better and improve with the increase in venture capital's participation degree [4, 21]; the other concludes that venture capital shareholding not only fails to improve the M&A performance of GEM-listed companies significantly but also adversely affects their short-term M&A performance [22]. Given this, this paper takes 468 M&A events accomplished by Chinese GEM-listed companies as the main merging party during 2014 and 2016 as research subjects to investigate the mechanism and effects of venture capital shareholding on GEM-listed enterprises' M&A performance; if a significant effect is found, further research will be conducted on the impact of shareholding ratio and venture capital syndicate on M&A performance. This research contributes to systematically clarifying the mechanism and impact of venture capital shareholding on the M&A performance of GEM-listed companies, addressing shortcomings in existing studies, and providing valuable insights for GEM-listed companies to understand and effectively utilize the active roles of venture capital shareholders in M&A activities.

## 2. Theoretical analysis and research hypothesis

### 2.1. The impact of venture capital shareholding on GEM-listed companies' M&A performance

During the external M&A process of GEM-listed companies, M&A target selection, M&A transaction process, and post-M&A integration are critical stages in determining their M&A performance. In this process, as institutional shareholders who own professional management capacity, experience in capital market operation, financial resources, and information network advantages, whether or not venture capital institutions can get the necessary room to function and own sufficient motivation to participate in it will substantially affect their performance of roles such as certification, monitoring, and value-added services, which in turn will affect the M&A performance of GEM-listed enterprises.

M&A target selection is a fundamental stage that affects GEM-listed enterprises' M&A performance. In this process [23], GEM-listed companies have to search for potential M&A targets, collect necessary information for them, assess and judge the strategic compatibility [24, 25] between the potential targets and themselves, and forecast possible M&A synergistic effects [26, 27]. Therefore, capacities of information gathering and assessing are two critical factors in determining the effectiveness of targeting, which are strengths of venture capital institutions, who can utilize their information network to seek viable M&A targets for the GEM-listed companies they invested and gather relevant information about the market, products, technology, human resources, and finance [21, 28]. In addition, venture capital shareholders can also utilize their professional abilities and capital market operating experience to provide technical support for assessing and judging the strategic compatibility of potential target enterprises and predicting possible M&A synergistic effects. Furthermore, when the management intends to select an unreasonable M&A target out of over-confidence [29] or to satisfy their selfish motives to establish a "managers' empire" [30], venture capital institutions can exercise their supervisory power over the major business decisions of the company through the board of directors to constrain the improper choices of management [31]. It can be seen that the performance of the monitoring and value-added services functions of venture capital is conducive to optimizing the M&A target selection for GEM-listed companies.

The M&A transaction process is the crucial stage in determining the M&A costs for GEM-listed companies. Venture capital's involvement in this process and its functions of certification, monitoring, and value-added services can help reduce the costs related to information collection, negotiation, and financing for GEM-listed companies and avoid unreasonable M&A premium payments. First, information support of venture capital organizations contributes to lowering the information collection and negotiation costs for GEM-listed companies. Mastering the necessary information related to M&A target enterprises is an important prerequisite for GEM-listed enterprises to make a reasonable valuation and formulate negotiation strategies. However, the systematic collection of information for M&A targets tends to generate high search costs [32]; therefore, venture capital institutions can help reduce it by utilizing their advantage of information networks to provide information support for GEM-listed enterprises. On this basis, the GEM-listed companies can make reasonable valuations of target enterprises and choose proper negotiation tactics based on sufficient information, which can help reduce significant disagreement between M&A parties, thereby shortening the negotiation process, improving the negotiation efficiency effectively and reducing the M&A negotiation costs for GEM-listed companies. Second, monitoring and value-added services provided by venture capital shareholders can assist in avoiding unreasonable payments of M&A premiums for GEM-listed companies. On the one hand, venture capital shareholders can utilize their professional capacities and experience in capital market operation to help GEM-listed

enterprises rationally analyze the potential synergies with targets and evaluate them reasonably; on the other hand, adequate supervision from venture capital shareholders can evade or minimize the adverse effects when management's blind arrogance may influence the M&A payment decisions of GEM-listed enterprises [33]. Third, functions of certification and value-added services of venture capital shareholders can help cut down the M&A financing costs of GEM-listed enterprises [13]. External financing is inevitable when GEM-listed enterprises have difficulties meeting their M&A payment demands with their funds. At this time, the financing cost becomes a crucial factor affecting the M&A performance. In the process of external financing, if there is apparent information asymmetry between enterprises and external capital providers, harsh terms may be set, and higher requirements may be raised for risk compensation out of their requests for risk defense which will increase the financing difficulty and expenses of GEM-listed companies [34, 35]. In practice, venture capital organizations have built extensive relationship networks with commercial banks, investment banks, fund companies, and other institutions owing to standing business transactions [36], which are conducive to disseminating and exchanging information. Therefore, the participation of venture capital shareholders in enterprises' M&A activities can spread messages related to M&A events widely within these networks, alleviating the information asymmetry between enterprises and outside capital providers, improving the financing environment, and decreasing companies' financing costs.

Post-M&A integration is a key step in determining the value creation of M&A for GEM-listed companies [37]. In this process, the participation of venture capital shareholders is favourable for accelerating the integration process and enhancing the effectiveness of integration for GEM-listed enterprises, thus increasing M&A earnings. According to the reputation mechanism theory and the information transmission theory, businesses owned by venture capital institutions with professional management teams and the advantages of resources and information networks are more likely to attract stakeholders' attention and earn their trust [13]. Therefore, venture capital's investment and involvement in the M&A transactions of GEM-listed enterprises are beneficial to boosting the faith of the management and staff of the acquired enterprises and reducing the resistance or negative response from the management and core workforce, thus effectively expediting the integration process. In addition, in the process of M&A integration, venture capital shareholders can use their professional abilities and experience to assist GEM-listed enterprises in formulating effective M&A integration schemes. At the same time, effective supervision also contributes to these schemes' continuous optimization and prompt execution.

In summary, the active participation of venture capital shareholders and their functions of certification, monitoring, and value-added services in the M&A process of GEM-listed enterprises can help optimize the selection of M&A targets, cut down M&A transaction costs, and increase M&A integration effectiveness, enhancing the M&A performance of GEM-listed enterprises. Based on the information above, this paper puts forward the following research hypotheses.

H1a: GEM-listed companies with venture capital shareholding perform much better in terms of short-term M&A performance than those without;

H1b: GEM-listed companies with venture capital shareholding perform much better in terms of long-term M&A performance than those without.

## 2.2. The impact of venture capital participation degree on GEM-listed companies' M&A performance

If venture capital shareholding has an obvious influence on the M&A performance of GEM-listed enterprises, it is necessary to pay further attention to the impact of venture capital

participation degree. The degree of venture capital participation is mainly reflected in two aspects: the proportion of venture capital shareholding and the existence of a venture capital syndicate, which refers to the simultaneous shareholding of one GEM-listed enterprise by two or more venture capital organizations. Under different degrees of participation, the differences in venture capital capacity, diversity of experience, financial resources, information advantages, and participation impetus will lead to different contributions to GEM-listed companies. Therefore different M&A performances will be generated.

Venture capitalists will be more motivated to take part in GEM-listed companies' M&A activities as a result of the growing venture capital shareholding ratio, which strengthens their interest bonds [21, 38]. Simultaneously, the increase in the shareholding ratio lifts the influence and discourse power of venture capital shareholders on GEM-listed enterprises, which is more conducive to utilizing professional management abilities and capital market operating experience to give full play to functions of monitoring and value-added services. A high venture capital shareholding ratio also transmits signals to external stakeholders that the company is of high quality [28, 32]. For M&A target enterprises, a higher venture capital shareholding ratio in the main merging party signifies a greater appreciation of the invested firm's worth, implying a higher quality and degree of trustworthiness of the company. The M&A performance of enterprises is promoted as a result of the fulfillment of certification, monitoring, and value-added services functions of venture capital shareholders, which helps the GEM-listed companies complete the selection of target enterprises, the process of M&A, and the post-M&A integration. Based on the above analysis, this paper proposes the following hypotheses.

H2a: There is a remarkably positive correlation between the venture capital shareholding ratio and GEM-listed companies' short-term M&A performance;

H2b: There is a remarkably positive correlation between the venture capital shareholding ratio and GEM-listed companies' long-term M&A performance.

Compared with single venture capital shareholding, joint venture capital investment can provide companies with complementary professional expertise [39] and management experience, which can complement strengths, pool ideas, and monitor each other as they play roles in monitoring and value-added services [40–42]. At the same time, joint venture capital investment can embed GEM-listed companies in a more extensive network of resources and relationships, providing timely access to the relevant resources and information needed [43]. In addition, joint venture capital investment is more conducive to the performance of certification functions, further strengthening the recognition and confidence of the merged and acquired companies to the GEM-listed enterprises. Based on the above analysis, the following hypotheses are proposed.

H3a: GEM-listed enterprises with joint venture capital investment significantly outperform those with single venture capital shareholding in terms of short-term M&A performance;

H3b: GEM-listed enterprises with joint venture capital investment significantly outperform those with single venture capital shareholding in terms of long-term M&A performance.

## 3. Research design

### 3.1. Sample selection and data sources

This paper takes M&A events announced between 2014 and 2016, whose main merging parties are GEM-listed enterprises and whose transactions were completed successfully as initial samples, and screens them using the following principles: (1) Multiple M&A events completed by

a single company in different years are considered as multiple different sample events; (2) If a company completes multiple M&A activities in the same year, the M&A event with the largest transaction amount will be selected; (3) If a single company makes multiple M&A announcements on the same day, they will be treated as one M&A event if the merged and acquired companies are related to each other. Otherwise, each will be treated as a separate M&A event; (4) M&A events in which financial GEM-listed companies are the main merger parties are removed; (5) M&A events with missing data and incomplete information are removed. After screening based on the above criteria, the paper finally obtained 468 sample observations.

The M&A samples required in this paper are mainly derived from the CSMAR Merger&Acquisition and Asset Restructuring Database. Companies' shareholders and other financial information are mainly collected from the CSMAR and RESSET databases. Information related to venture capital institutions is mainly gathered from the WIND database and the Zero2IPO database. Relevant data that is undisclosed or missing from the above databases is manually collected and organized from Sina Finance Web and the official websites of related companies.

### 3.2. Variables design

Based on the preceding theoretical analysis and research hypotheses, this paper formulates the following theoretical model (Fig 1) and devises pertinent variables.

**3.2.1. Explained variables.** According to previous research hypotheses, this paper constructs short-term M&A performance and long-term M&A performance indicators to measure the M&A performance of GEM-listed companies.

*3.2.1.1. Short-term M&A performance.* Domestic and foreign scholars often use the event study method to measure companies' short-term M&A performance. This paper also follows this practice and adopts this method to measure the short-term M&A performance of GEM-listed companies. Two aspects will be taken into consideration concerning the selection of the event window. On the one hand, the event window should not be too small otherwise share price fluctuation caused by early market entry due to M&A news may be missed. On the other hand, information that is irrelevant to the M&A event may be involved if the event window is set too long. Therefore, the cumulative abnormal return of GEM-listed enterprises stocks in the five days before and after the first announcement date of M&A, i.e., CAR (-5, 5), is used to evaluate the short-term M&A performance, while CAR (-10, 10) is used as the robustness test indicators.

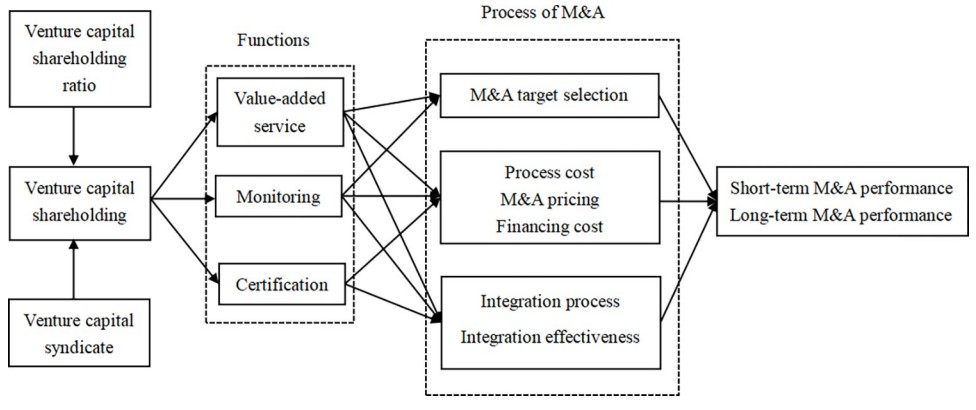

**Fig 1. Theoretical model.**

*3.2.1.2. Long-term M&A performance.* This paper uses the financial index method to measure GEM-listed enterprises' long-term M&A performance. As long-term M&A performance will be reflected in various economic activities of enterprises and no single indicator can fully capture the economic effects of M&A [44], this paper argues that it is more reasonable to use multiple financial indicators to measure enterprises' long-term M&A performance comprehensively. Specific implementation methods are as follows: (1) Select corresponding financial indicators with the guidance of comprehensiveness and materiality; (2) Conduct factor analyses on various standardized financial indicators of each sample enterprise for one year prior to M&A, one year after M&A, and three years after M&A and extract common factors; (3) Calculate the score of each common factor on each sample, and then multiply it with the weight generated by the percentage of the variance contribution rate of each common factor to the cumulative variance contribution rate, to obtain the composite score of the performance of each sample company for one year before the M&A, one year after the M&A, and three years after the M&A, denoting them by $F_i^{-1}$、 $F_i^1$、 $F_i^3$, respectively; (4) $F_i^1 - F_i^{-1}$ is used as sample companies' long-term M&A performance (simplified as F1-F-1) and $F_i^3 - F_i^{-1}$ is used as the robustness test index(simplified as $F_3$-$F_{-1}$).

**3.2.2. Explaining variables.** Based on the preceding research hypotheses, the following explaining variables are set in this paper.

*3.2.2.1. Venture capital (VC).* Drawing on the practice of Yang et al. [45], sample enterprises are screened successively by the following steps: searching and analyzing the names of the top ten shareholders of the GEM-listed enterprises, querying the directory of venture capital institutions and the directory of private equity investment institutions, and querying the business scope of shareholders in these institutions. If the sample enterprise has venture capital shareholding, the value of VC is 1; otherwise, the value is 0.

*3.2.2.2. Venture capital shareholding share (VCshare).* Take the sum of the percentage of shares held by venture capital organizations among the top ten shareholders of GEM-listed enterprises in the year when the M&A event occurs as the venture capital shareholding ratio.

*3.2.2.3. Venture capital syndicate (VCsynd).* According to whether or not their top ten shareholders contain two or more venture capital institutions in the year when the M&A event occurs, GEM-listed enterprises with venture capital investment are categorized into enterprises with venture capital syndicate and single venture capital investment enterprises. If a sample enterprise is a venture capital syndicate, the value of VCsynd is 1; otherwise, the value is 0.

**3.2.3. Control variables.** In this paper, the following control variables have been used: relative size of M&A, M&A type, M&A payment method, company size, company growth capacity, liability level, cashflow, ownership concentration, earning capacity, operation capacity, and company age. Additionally, industry and year variables are also under control. Variables names, codes, and measurements involved in this paper are presented in Table 1.

## 3.3. Construction of models

Based on the aforementioned variable design, in order to test hypotheses H1a and H1b, the following multiple linear regression Model 1 and Model 2 are constructed in this paper:

$$CAR(-5,5) = \alpha + \beta_1 VC + \beta_2 RS + \beta_3 Type + \beta_4 PT + \beta_5 Size + \beta_6 Growth + \beta_7 Lev + \beta_8 Cash$$
$$+ \beta_9 TOP5 + \beta_{10} Roa + \beta_{11} Tat + \beta_{12} Age + \sum Ind + \sum Year + \varepsilon \qquad (1)$$

**Table 1. Variables definitions.**

| Variable | Variable name | Variable code | Measurement | Data source |
|---|---|---|---|---|
| Explained variables | Short-term M&A performance | CAR(-5, 5) | Cumulative abnormal returns of stock for the five days before and after the date of the M&A announcement | CSMAR Database RESSET Database |
| | Long-term M&A performance | $F_1$-$F_{-1}$ | The difference in comprehensive score of performance in the year before and after the date of the M&A announcement | |
| Explaining variables | Venture capital shareholding | VC | If there is venture capital shareholding among the top ten shareholders, the value is 1; otherwise, it is 0 | CSMAR Database WIND Database Zero2IPO Database |
| | Venture capital shareholding ratio | VCshares | The sum of the shareholding ratio of venture capital institutions among the top ten shareholders | |
| | Venture capital syndicate | VCsynd | If the top ten stockholders include two or more venture capital firms, the value is 1; otherwise, it is 0 | |
| Control variables | Relative size | RS | M&A transaction amount/total assets in the year before the M&A | CSMAR Database WIND Database RESSET Database |
| | M&A type | Type | Take 1 for relevant M&A, 0 for others | |
| | Payment terms | PT | Take 1 for cash payment, 0 for others | |
| | Company size | Size | Natural logarithm of total assets in the year before the M&A | |
| | Company growth | Growth | The growth rate of operating income in the year before the M&A | |
| | Liability level | Lev | Asset-liability ratio in the year before the M&A | |
| | Cashflow | Cash | FCF in the year before the M&A/total assets in the year before the M&A | |
| | Shareholding concentration | TOP5 | The shareholding ratio of the top five shareholders in the year before the M&A | |
| | Earning capacity | Roa | Return on total assets in the year before the M&A | |
| | Operation capacity | Tat | Total asset turnover in the year before the M&A | |
| | Age of company | Age | Natural logarithm of the number of years from the establishment of the company to the year of M&A | |
| | Industry dummy variable | Ind | The samples are divided into 14 industries, and 13 industry dummy variables are set. | Guidelines on Industry Classification of Listed Companies in China [46] |
| | Year dummy variable | Year | The sample data is collected from three years, and two year dummy variables are set. | |

$$F_1 - F_{-1} = \alpha + \beta_1 VC + \beta_2 RS + \beta_3 Type + \beta_4 PT + \beta_5 Size + \beta_6 Growth + \beta_7 Lev + \beta_8 Cash$$
$$+ \beta_9 TOP5 + \beta_{10} Roa + \beta_{11} Tat + \beta_{12} Age + \sum Ind + \sum Year + \varepsilon \qquad (2)$$

Furthermore, based on samples with venture capital shareholding, in order to test hypotheses H2a, H2b, H3a, and H3b, the following multiple linear regression Model 3, Model 4, Model 5, and Model 6 are constructed in this paper:

$$CAR(-5, 5) = \alpha + \beta_1 VCshares + \beta_2 RS + \beta_3 Type + \beta_4 PT + \beta_5 Size + \beta_6 Growth + \beta_7 Lev + \beta_8 Cash$$
$$+ \beta_9 TOP5 + \beta_{10} Roa + \beta_{11} Tat + \beta_{12} Age + \sum Ind + \sum Year + \varepsilon \qquad (3)$$

$$F_1 - F_{-1} = \alpha + \beta_1 VCshares + \beta_2 RS + \beta_3 Type + \beta_4 PT + \beta_5 Size + \beta_6 Growth + \beta_7 Lev$$
$$+ \beta_8 Cash + \beta_9 TOP5 + \beta_{10} Roa + \beta_{11} Tat + \beta_{12} Age + \sum Ind + \sum Year + \varepsilon \qquad (4)$$

$$CAR(-5, 5) = \alpha + \beta_1 VCsynd + \beta_2 RS + \beta_3 Type + \beta_4 PT + \beta_5 Size + \beta_6 Growth + \beta_7 Lev + \beta_8 Cash$$
$$+ \beta_9 TOP5 + \beta_{10} Roa + \beta_{11} Tat + \beta_{12} Age + \sum Ind + \sum Year + \varepsilon \qquad (5)$$

$$F_1 - F_{-1} = \alpha + \beta_1 VCsynd + \beta_2 RS + \beta_3 Type + \beta_4 PT + \beta_5 Size + \beta_6 Growth + \beta_7 Lev + \beta_8 Cash$$
$$+ \beta_9 TOP5 + \beta_{10} Roa + \beta_{11} Tat + \beta_{12} Age + \sum Ind + \sum Year + \varepsilon \qquad (6)$$

# 4. Empirical results and analysis

## 4.1. Descriptive statistics of variables

In order to better understand each variable, descriptive statistics were conducted based on the full sample, and the results are presented in Table 2.

As can be seen from Table 2, the mean values of CAR(-5, 5) and CAR(-10, 10) for short-term M&A performance are 9.71% and 9.29%, respectively, indicating that the stock market shows a positive reaction to the announcement of the M&A; the mean values of $F_1$-$F_{-1}$ and $F_3$-$F_{-1}$ for long-term M&A performance are positive but close to 0, suggesting that the M&A activities have very limited impact on enhancement of corporate value in the long run. In terms of explaining variables, 52.56% of the sample enterprises have venture capital shareholding, and 38.21% have joint venture capital investment, indicating that GEM-listed companies have a higher prevalence of venture capital shareholding and venture capital syndicate. In addition, the mean value of venture capital shareholding ratio is 14.36%, but the share highly varies among different enterprises. As for the control variables, the mean value of the M&A relative size is 39.41%, and 85.26% of the M&A samples are relevant M&A, illustrating that the M&A relative size of GEM-listed enterprises is large and most M&A samples belong to relevant M&A; the mean value of the top five shareholders' ownership proportion in sample companies is 58%, suggesting that the ownership concentration of GEM-listed enterprises is generally high.

**Table 2. Descriptive statistics for the full sample.**

| Variable | Number of samples | Mean | SD | Minimum | Maximum |
|---|---|---|---|---|---|
| CAR(-5,5) | 468 | 0.0971 | 0.2401 | -0.7124 | 0.7379 |
| CAR(-10,10) | 468 | 0.0929 | 0.2965 | -1.2249 | 1.2329 |
| $F_1$-$F_{-1}$ | 468 | 1.0177E-16 | 0.5178 | -2.6705 | 2.5095 |
| $F_3$-$F_{-1}$ | 468 | 1.73176E-17 | 0.5730 | -3.4898 | 2.0450 |
| VC | 468 | 0.5256 | 0.4999 | 0.0000 | 1.0000 |
| VCshares | 246 | 14.3565 | 16.4048 | 0.4200 | 69.6000 |
| VCsynd | 246 | 0.3821 | 0.4869 | 0.0000 | 1.0000 |
| RS | 468 | 0.3941 | 0.6173 | 0.0000 | 6.2069 |
| Type | 468 | 0.8526 | 0.3549 | 0.0000 | 1.0000 |
| PT | 468 | 0.5427 | 0.4987 | 0.0000 | 1.0000 |
| Size | 468 | 20.9353 | 0.6512 | 19.3845 | 23.1543 |
| Growth | 468 | 26.1101 | 36.2457 | -61.8269 | 386.5970 |
| Lev | 468 | 25.8990 | 15.3226 | 1.1034 | 88.6428 |
| Cash | 468 | -0.0755 | 0.1992 | -1.9555 | 0.2005 |
| Top5 | 468 | 57.0863 | 11.9582 | 26.0844 | 86.4952 |
| Roa | 468 | 6.4706 | 5.5825 | -32.5666 | 27.5049 |
| Tat | 468 | 0.5315 | 0.3773 | 0.0623 | 6.2913 |
| Age | 468 | 2.4875 | 0.3582 | 1.3863 | 3.2581 |

## 4.2. Regression analysis of venture capital shareholding's effect on GEM-listed enterprises' M&A performance

Based on the full sample, regression analyses are conducted for Model 1 and Model 2, respectively, and the results are listed in Column (1) and Column (4) of Table 3. It can be inferred from Column (1) that there is a significantly positive correlation between venture capital shareholding and the short-term M&A performance of GEM-listed companies at the 5% significance level, that is, the short-term M&A performance of GEM-listed companies with venture capital investment is noticeably superior to those without. Hypothesis H1a in this paper has been verified. Concerning control variables, there is a remarkably positive correlation

**Table 3. Regression results of the venture capital shareholding's impact on GEM-listed enterprises' M&A performance.**

| | Short-term M&A Performance CAR(-5,5) | | | Long-term M&A Performance $F_1$-$F_{-1}$ | | |
|---|---|---|---|---|---|---|
| | (1) | (2) | (3) | (4) | (5) | (6) |
| VC | 0.0461** | | | 0.104** | | |
| | (2.11) | | | (2.33) | | |
| VCshares | | 0.00232** | | | -0.000586 | |
| | | (2.51) | | | (-0.31) | |
| VCsynd | | | 0.0669** | | | -0.0996 |
| | | | (2.20) | | | (-1.60) |
| RS | 0.0729*** | 0.102*** | 0.0969*** | -0.0307 | -0.0321 | -0.0241 |
| | (3.41) | (3.51) | (3.31) | (-0.70) | (-0.53) | (-0.40) |
| Type | -0.0353 | -0.0219 | -0.0316 | -0.0714 | -0.00193 | 0.00841 |
| | (-1.12) | (-0.50) | (-0.72) | (-1.11) | (-0.02) | (0.09) |
| PT | -0.0839*** | -0.0613* | -0.0528 | -0.120** | -0.0668 | -0.0780 |
| | (-3.31) | (-1.69) | (-1.44) | (-2.32) | (-0.89) | (-1.04) |
| Size | 0.00512 | -0.00388 | -0.00305 | -0.0688 | -0.0385 | -0.0431 |
| | (0.25) | (-0.14) | (-0.11) | (-1.64) | (-0.65) | (-0.73) |
| Growth | -0.000573 | -0.000752 | -0.000748 | -0.00237*** | -0.000896 | -0.000744 |
| | (-1.57) | (-1.34) | (-1.33) | (-3.19) | (-0.77) | (-0.64) |
| Lev | -0.00145* | -0.000216 | -0.000473 | -0.000626 | -0.00254 | -0.00220 |
| | (-1.71) | (-0.18) | (-0.39) | (-0.36) | (-1.01) | (-0.88) |
| Cash | 0.0886 | 0.0884 | 0.0904 | 0.333*** | 0.350** | 0.340** |
| | (1.54) | (1.25) | (1.27) | (2.85) | (2.40) | (2.34) |
| Top5 | -0.00112 | -0.00110 | -0.000721 | -0.000497 | -0.00141 | -0.00131 |
| | (-1.15) | (-0.81) | (-0.54) | (-0.25) | (-0.50) | (-0.47) |
| Roa | -0.000740 | -0.00190 | -0.00148 | -0.0236*** | -0.0297*** | -0.0303*** |
| | (-0.34) | (-0.58) | (-0.45) | (-5.33) | (-4.38) | (-4.48) |
| Tat | 0.0131 | -0.126* | -0.109 | -0.146** | 0.166 | 0.161 |
| | (0.37) | (-1.68) | (-1.45) | (-2.03) | (1.07) | (1.05) |
| Age | 0.0125 | -0.0142 | -0.00812 | 0.0523 | 0.0471 | 0.0386 |
| | (0.40) | (-0.34) | (-0.19) | (0.81) | (0.54) | (0.45) |
| Ind | Controlled | Controlled | Controlled | Controlled | Controlled | Controlled |
| Year | Controlled | Controlled | Controlled | Controlled | Controlled | Controlled |
| N | 468 | 246 | 246 | 468 | 246 | 246 |
| AdjR2 | 0.1065 | 0.1688 | 0.1633 | 0.2025 | 0.1212 | 0.1311 |

Note

***, **, and * indicate significance at the levels of 1%, 5%, and 10%, respectively, and the figures in parentheses are t values.

between M&A relative size and GEM-listed companies' short-term M&A performance at the 1% significance level. In contrast, the M&A payment method and the liability level negatively correlate with GEM-listed companies' short-term M&A performance at the 1% and 10% significance levels, respectively. This suggests that a large-scale M&A activity by GEM-listed enterprises is more likely to be positively responded to by the stock market, whereas investors are more negative towards M&A activities with cash payments and high liability levels.

It can be concluded from column (4) of Table 3 that there is a significantly positive relationship between venture capital shareholding and GEM-listed companies' long-term M&A performance at the 5% significance level, that is, the short-term M&A performance of GEM-listed companies with venture capital investment is noticeably superior to those without. Accordingly, hypothesis H1b is confirmed. Regarding control variables, there is a strikingly negative correlation between the long-term M&A performance and both the M&A payment method and operation capacity at the 5% significance level. At the same time, both the growth and earning capacity of GEM-listed enterprises negatively correlate with the long-term M&A performance at the 1% significance level, whereas cashflow positively correlates with the long-term M&A performance at the 1% significance level. This result shows that the non-cash payment method and relatively abundant cash flow are conducive to improving the long-term M&A performance of GEM-listed enterprises, while the more prominent the operation capacity, earning capacity, and growth of GEM-listed enterprises are prior to M&A, the more obvious the adverse impact on enterprise performance is one year later.

## 4.3. Regression analysis of the effect of venture capital participation degree on GEM-listed enterprises' M&A performance

Considering the significant impact of venture capital shareholding on GEM-listed companies' M&A performance, this paper further conducts regression analyses of Models 3 to Model 6 based on grouped samples, and the results are shown in Columns (2), (5), (3), and (6) of Table 3, respectively. It can be seen from Column (2) in Table 3 that a significantly positive relationship exists between venture capital shareholding ratio and the short-term M&A performance at the 5% significance level; that is, an increase in venture capital shareholding ratio contributes to enhancing GEM-listed enterprises' short-term M&A performance, verifying hypothesis H2a of this paper. Column (3) of Table 3 concludes that there is a significantly positive relationship between venture capital syndicate and GEM-listed enterprises' short-term M&A performance at the 5% significance level, suggesting that the short-term M&A performance of GEM-listed enterprises with venture capital syndicate remarkably excesses that of those with single venture capital shareholding. Hence, hypothesis H3a of this paper is validated. Moreover, in the results of two regression analyses, there is a significantly positive relationship between M&A relative size and GEM-listed companies' short-term M&A performance at the 1% significance level, which confirms once again that M&A activities with larger size by GEM-listed enterprises are more likely to be positively responded to by the stock market.

Column (5) in Table 3 reveals that there is an insignificant, negative relationship between the venture capital shareholding ratio and the long-term M&A performance of GEM-listed enterprises, showing that an increase in venture capital shareholding does not considerably improve GEM-listed enterprises' long-term M&A performance. Therefore, hypothesis H2b in this paper is invalid. It can be inferred from Column (6) in Table 3 that there is an insignificantly negative relationship between venture capital syndicate and GEM-listed companies' long-term M&A performance, indicating that there is little remarkable difference between the effect of a venture capital syndicate and a single venture capital shareholding on GEM-listed

companies' long-term M&A performance. Hypothesis H3b in this research is untenable. Moreover, in both regressions, there is a significantly positive relationship between cashflow and the long-term M&A performance of GEM-listed enterprises at the 5% significance level and a significantly negative relationship between earning capacity and long-term M&A performance at the 1% significance level, demonstrating once more that owning abundant cashflows is conducive to promoting GEM-listed enterprises' long-term M&A performance. At the same time, the more exceptional the earning capacity of GEM-listed enterprises is before M&A, the more evident the negative effects on the company's performance are one year later.

## 4.4. Robustness test

To confirm the reliability of the above statistics, we take CAR(-10,10) as the substitutable explained variable to measure GEM-listed enterprises' short-term M&A performance and insert it into Model 1, Model 3, and Model 5, respectively. At the same time, the difference in comprehensive scores on performance between three years after M&A and one year before M&A, i.e., $F_3$-$F_{-1}$, is taken as the substitutable explained variable to measure the long-term M&A performance and inserted into Model 2, Model 4, and Model 6, respectively. Then, regression analyses are conducted on Model 1 and Model 2 based on full samples and Model 3, Model 4, Medel 5, and Model 6 based on grouped samples. The robustness test results manifest that there is no substantial difference in other material regression results except a significantly positive correlation between venture capital shareholding and GEM-listed enterprises' long-term M&A performance at the 10% significance level. The reliability of previous empirical results is verified. Due to space limitations, the robustness test results are not presented in detail, but they are available if required.

## 4.5. Further analysis based on the empirical results

The regression result in the first stage shows that GEM-listed companies with venture capital shareholding outperform those without in both the short-term and long-term M&A performance. This result verifies the previous analysis in this paper, that is, through playing roles in the certification, monitoring, and value-added services functions, venture capital shareholders are conducive to optimizing the selection of M&A targets, reducing M&A transaction costs and enhancing integration effectiveness, thus improving GEM-listed enterprises' M&A performance. The regression result in the second stage suggests that an increase in venture capital participation degree contributes to improving GEM-listed enterprises' short-term M&A performance but makes little contribution to long-term M&A performance enhancement. In view of this result, this paper attempts to make a further analysis from the perspective of the functioning of venture capital shareholders.

An increase in the shareholding ratio of venture capital shareholders implies corresponding growth in their influence on GEM-listed enterprises, enabling them to play their roles in certification, monitoring, and value-added services more effectively. Moreover, the rise in the number of venture capital shareholders also denotes that a more reliable certification function and complementary monitoring and value-added services functions can be performed on GEM-listed enterprises. This mechanism aligns with investors' basic expectations and can be embodied by the share price fluctuation. In the external M&A process of GEM-listed enterprises, the conditions to fulfill the responsibilities of venture capital shareholders are different. Shareholding is the best illustration for their certification function and does not require too many preconditions. However, the fulfillment of their monitoring and value-added service functions is not quite the same. Under the conventional principal-agent governance structure, it is often the controlling shareholders of the enterprise or those acting in concert with relative

shareholding advantages who can exert substantial influence on the major business decisions of the enterprise. If there is no major change in the shareholding structure of the enterprise, an appropriate increase in the shareholding ratio of individual shareholders cannot accordingly increase their actual influence on the enterprise. Most GEM-listed enterprises have a one-share ownership structure, and the founder shareholders or their concerted action people often own the absolute controlling rights over the enterprise. Under this shareholding structure, it is difficult to enhance the substantial influence of venture capital shareholders on the major operational decisions by moderately increasing the proportion of their single or joint shareholdings. Therefore, in the M&A-related decision-making process of GEM-listed enterprises, the monitoring function performance of venture capital shareholders is difficult to align with the increase in their shareholding ratio. Additionally, for venture capital shareholders, fulfilling the value-added services function usually means the input of resources and energies and the occurrence of corresponding costs. Therefore, venture capital shareholders' motivation and commitment to provide value-added services will be influenced by a lack of attention and recognition from controlling shareholders or the existence of "free riding" psychology [47] among venture capital shareholders. To sum up, the performance of monitoring and value-added services functions of venture capital shareholders is inconsistent with the increase in their participation degree, thus failing to significantly improve GEM-listed enterprises' long-term M&A performance.

## 5. Conclusion

Given the scarcity of existing studies and contrasting research findings, this paper takes 468 M&A events completed by the acquirer of Chinese GEM-listed companies between 2014 and 2016 as research samples to investigate the impact of venture capital shareholding on GEM-listed companies' M&A performance. The empirical results show (1) GEM-listed companies that have venture capital shareholding perform much better in short-term and long-term M&A performance than those that do not; (2) An increase in venture capital shareholding ratio significantly improves GEM companies' short-term M&A performance, but not the long-term M&A performance; (3) Venture capital syndicates can remarkably enhance GEM-listed enterprises' short-term M&A performance, but contribute little to the growth of long-term M&A performance. Based on the empirical results, further analysis in this paper concludes that the prevalent one-share shareholding structure, inadequate attention and recognition from controlling shareholders, and the "free riding" mentality among part of the venture capital shareholders will substantially influence their performance of monitoring and value-added services functions. Therefore, in order to make venture capital shareholders better play roles in improving GEM-listed enterprises' long-term M&A performance, GEM-listed companies can design more flexible decision-making mechanisms for M&A activities and, at the same time, attach more importance to and motivate venture capital shareholders to play value-added services function. Venture capital shareholders also need to get rid of the interruption of "free riding" thinking to play monitoring and value-added services functions positively and actively.

## 6. Discussion

In the current relevant studies with a limited number, two incompatible research results are presented regarding the impact of venture capital shareholding on the M&A performance of GEM-listed companies: One conclusion suggests that both short-term and long-term M&A performance of GEM-listed companies with venture capital shareholding are superior; furthermore, an increase in venture capital shareholding markedly enhances the M&A performance

of GEM-listed companies. The other research conclusion indicates that not only did venture capital shareholding fail to improve the long-term M&A performance of GEM-listed companies notably, but it also negatively affected their short-term M&A performance. This study provides a comprehensive analysis of the impact of venture capital institutions' functions, such as certification, monitoring, and value-added services, on the crucial stages in the M&A process of GEM-listed companies and systematically clarifies the mechanism of venture capital shareholding on the M&A performance of GEM-listed companies. The empirical analysis based on this not only verifies the positive effect of venture capital shareholding on the short-term and long-term M&A performance of GEM-listed companies but also further explains the different effects of an increase in venture capital shareholding and a venture capital syndicate on the short-term and long-term M&A performance of GEM-listed companies. Compared with existing studies, this research more systematically demonstrates the mechanism by which venture capital shareholding affects the M&A performance of GEM-listed companies and analyzes the impact of venture capital shareholding on the short-term and long-term M&A performance of GEM-listed companies at a deeper level, addressing the shortcomings in current studies and providing a valuable reference for these companies to rationally understand and effectively utilize the positive effects of venture capital shareholders in M&A activities.

## 7. Limitation

Although this paper explored the impact of venture capital shareholding on the M&A performance of GEM-listed companies from multiple perspectives, there are still several areas for improvement: (1) The capacity differences among venture capital institutions are objective. Limited by the studying materials, the specific impact of capacity difference among venture capital institutions on the M&A performance of GEM-listed companies is not further analyzed;(2) This study analyzed the impact of venture capital shareholding and shareholding share on the M&A performance of GEM-listed companies. However, limited to the completeness of data, it fails to conduct further a classification study on the nature of property rights, shareholding ratio, shareholding period, and other factors of venture capital institutions; (3) This research is carried out from the perspective of GEM-listed companies as the acquirer, failing to pay attention to the value changes of target company after M&A. All deficiencies listed above are research directions in the future.

## Acknowledgments

All authors wish to appreciate the valuable comments from the anonymous reviewers. All errors remain the sole responsibility of the authors.

## Author Contributions

**Conceptualization:** Kun Chao.

**Data curation:** Kun Chao, Meijia Wang, Yanyong Hu.

**Formal analysis:** Kun Chao, Yanyong Hu.

**Funding acquisition:** Yanyong Hu.

**Investigation:** Meijia Wang, Shixue Wang.

**Methodology:** Kun Chao, Yanyong Hu.

**Project administration:** Kun Chao, Yanyong Hu.

**Software:** Meijia Wang, Shixue Wang.

**Supervision:** Meijia Wang, Shixue Wang.

**Validation:** Kun Chao, Yanyong Hu.

**Writing – original draft:** Kun Chao.

**Writing – review & editing:** Meijia Wang, Yanyong Hu.

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
