## [Decision Letter · Decision Letter 0]

8 Apr 2024

PONE-D-24-04863Can venture capital shareholding improve M&A performance? An empirical study based on Chinese GEM-listed companiesPLOS ONE

Dear Dr. Hu,

In view of the referees’ feedback and my own reading of your paper, we invite you to address all issues noted below. We consider these issues to be major in nature, requiring more than a superficial or minor revision. We have particular concerns about the lack of a clear research question, the robustness of the methods and analysis and the clarity of the narrative and expression.

We are confident that the issues identified could be resolved with a major revision, so we invite you to address the issues noted below and resubmit the manuscript for a new revision round.

We look forward to receiving your revised manuscript.

Kind regards,

Juan E. Trinidad Segovia, PhD

Section Editor

PLOS ONE

Journal Requirements:

"This research was funded by the Basic Research Funds of China University of Mining and Technology (Beijing) - Fund for Cultivation of Top notch Innovative Talents for Doctoral Graduates (No. BBJ2023046)."

Reviewers' comments:

Reviewer's Responses to Questions

**Comments to the Author**

1. Is the manuscript technically sound, and do the data support the conclusions?

Reviewer #1: Yes

Reviewer #2: Yes

2. Has the statistical analysis been performed appropriately and rigorously? 

Reviewer #1: Yes

Reviewer #2: Yes

3. Have the authors made all data underlying the findings in their manuscript fully available?

Reviewer #1: Yes

Reviewer #2: Yes

4. Is the manuscript presented in an intelligible fashion and written in standard English?

Reviewer #1: No

Reviewer #2: Yes

5. Review Comments to the Author

Reviewer #1: Dear Authors and Editor,

Firstly, thank you for submitting your paper “Can venture capital shareholding improve M&A performance? An empirical study based on Chinese GEM-listed companies” (PONE-D-24-04863) to Plos One. After reading, I should point out these issues. The following is from my comments.

1. The authors should check the language throughout the whole manuscript.

2. The authors should add background, purpose, and value section into abstract.

3. In the introduction and literature review, the authors should pay more attention to the recent and important literature. Specially, the year of 2022-2024. The following upon the topic “venture capital shareholding” can be cited, I suggest that

Doi: 10.1108/EJIM-03-2021-0161

Doi: 10.1016/j.heliyon.2023.e13192

4. I can not see the value and contribution of the study in the end of introduction. The authors should improve the quality of introduction.

5.The data source is unclear and the quality of methodology is also poor.

6. I can not see the fig. of the research model. The authors should add it before the section of “3.2. Variables design”.

7. The authors should add the source into “Table 1. Variables definitions”.

8. The authors should add the section of “discussion”. Meanwhile, the authors should add more to the “Discussion section”, meanwhile, comparing the result with previous ones. The authors should add the discussion section. At the present version, it is missing. It is illustrate he value and contribution of the manuscript.

9. The authors should check the important and recent ones.

10. The authors should add limitation section.

12. The authors should check all the references’ formats based on Plos One’s style and the present version lacks DOIs.

13. The authors should strictly address these comments please.

To summarize, I should choose “Major Revision” and welcome the revised submission in the future. Good luck !

All the best

Mar 17, 2024

Reviewer #2: Based on a sample of 468 M&A activities completed by M&A parties of Chinese GEM-listed firms from 2014 to 2016, this paper explores the impact of venture capital shareholding on the M&A performance of GEM-listed firms. On this basis, the differences between venture capital shareholding and joint venture investment on short-term and long-term M&A performance are explored separately. And the reasons for the differences between venture capital shareholding ratio and joint venture investment on short-term and long-term investment are explained based on the perspective of the operation of venture capital shareholders. The topic selection is novel, the argument is sufficient, and the detailed reasons are explained for the conclusions of violating the assumptions. However, the following points need to be noted: 1. please explain why the sample selection time node is 2014-2016

2. the theoretical and practical significance of the study is not sufficiently elaborated, and it is suggested to add appropriately

6. PLOS authors have the option to publish the peer review history of their article (what does this mean?). If published, this will include your full peer review and any attached files.

Reviewer #1: No

Reviewer #2: **Yes: **Xiaofei Shi

---

## [Author Response · Author response to Decision Letter 0]

11 Jun 2024

Response to the comments

Dear Editor,

We sincerely thank the editor and two reviewers for their valuable feedback that we have used to improve the quality of our manuscript entitled " Can venture capital shareholding improve M&A performance? An empirical study based on Chinese GEM-listed companies " (PONE-D-24-04863). The reviewer comments are laid out below in Bold font. We corrected spelling and grammar errors and polish the whole manuscript through a professional language editing institution. In this revised version, changes to our manuscript were all in italic within the document. We have carefully addressed the comments raised by the reviewers in a point-to-point manner. Response to the reviewers’ comments is presented as follows:

Response to Reviewer 1:

Dear Authors and Editor,

Firstly, thank you for submitting your paper “Can venture capital shareholding improve M&A performance? An empirical study based on Chinese GEM-listed companies” (PONE-D-24-04863) to Plos One. After reading, I should point out these issues. The following is from my comments.

Response: We would like to thank you for your careful reading, helpful comments, and constructive suggestions, which have significantly improved the presentation of our manuscript. We have carefully considered all comments from the reviewers and answered all comments accordingly in our manuscript.

1. The authors should check the language throughout the whole manuscript.

Response: Thanks for your careful reading. We have re-examined and polished the language of the whole manuscript and corrected spelling mistakes, improper expressions and grammatical errors to make it more readable. 

2. The authors should add background, purpose, and value section into abstract.

Response: We would like to thank you for your careful reading, helpful comments, and constructive suggestions, which have significantly improved the presentation of our manuscript. According to your suggestion, we have supplemented this study’s background, purpose, and value section into abstract. The revised abstract is as follows: 

Existing studies have explored the impact of venture capital shareholding on the GEM-listed companies before and after listing from multiple perspectives. However, there has been limited research on the influence of venture capital shareholding on these companies’ mergers and acquisitions(M&A) activities and performance. Additionally, two conflicting research findings have been presented in limited relevant studies. In order to clarify the mechanism by which venture capital shareholding affects M&A activities and performance of GEM-listed companies and verify existing research conclusions, this paper takes 468 M&A events completed by the acquirer of Chin’s GEM-listed companies between 2014 and 2016 as samples to explore venture capital shareholding’s effects on the M&A performance of GEM-listed enterprises. The empirical findings demonstrate that GEM-listed enterprises with venture capital shareholding perform significantly better in terms of short-term and long-term M&A performance than those without; with the increase in venture capital shareholding ratio, the short-term M&A performance of GEM-listed enterprises has remarkably improved, but the long-term M&A performance does not show obvious correlation; joint investment of venture capital can significantly improve the short-term M&A performance of GEM-listed enterprises, but it has no substantial influence on long-term M&A performance. Based on further analysis of the empirical study, it is concluded that the common one-share ownership structure of GEM-listed enterprises is not conducive to the play of the monitoring function of venture capital, and the insufficient incentives and free-riding thinking also weaken the motivation and input of some venture capital shareholders to provide value-added services. This study systematically elucidates the mechanism and impact of venture capital shareholding on the M&A performance of GEM-listed companies, addressing the shortcomings in existing research. It is conducive for GEM-listed companies to gain a rational understanding and effectively leverage the active role of venture capital shareholders in M&A activities.

3. In the introduction and literature review, the authors should pay more attention to the recent and important literature. Specially, the year of 2022-2024. The following upon the topic “venture capital shareholding” can be cited, I suggest that Doi: 10.1108/EJIM-03-2021-0161 Doi: 10.1016/j.heliyon.2023.e13192.

Response: Thank you for your thoughtful comments. Yi et al. (2021) stated that venture capital can significantly promote open innovation of enterprises and this promoting effect is more significant when the venture capital institutions have profounder industry experience, higher shareholding ratio and are syndicated. Lyu et al.(2023) invstigated the impact of stakeholder network characteristics on a company’s open innovation performance. Therefore, these articles you suggested are relevant to the section below, and we have included it. 

Some scholars at home and abroad have researched the impact of venture capital on investing and financing behaviors [5], earnings management [7], establishing an efficient board of directors [8], improving corporate governance structures [9], implementing equity incentives and replacing CEO [10], technological innovation [11-14] after going public and other aspects.

Specific references are listed as follows:

11.Yi R, Wang H, Lyu B, Xia Q. Does venture capital help to promote open innovation practice? Evidence from China. Eur J Innov Manag. 2023;26(1):1-26. doi: 10.1108/EJIM-03-2021-0161.

12.Lyu B, Yi R, Fan G, Zhang Y. Stakeholder network for developing open innovation practice of China's manufacturing enterprises. Heliyon. 2023;9(3):e13192-e. doi: 10.1016/j.heliyon.2023.e13192.

4. I can not see the value and contribution of the study in the end of introduction. The authors should improve the quality of introduction.

Response: Thank you for your thoughtful comments. As a result of your suggestions, we have added expressions about the value and contribution of this study at the end of the introduction to make it high-quality. Specific expressions are as follows：

This research contributes to systematically clarifying the mechanism and impact of venture capital shareholding on the M&A performance of GEM-listed companies, addressing shortcomings in existing studies, and providing valuable insights for GEM-listed companies to understand and effectively utilize the active roles of venture capital shareholders in M&A activities.

5. The data source is unclear and the quality of methodology is also poor.

Response: We are very grateful to the reviewer for pointing out this concern. As a result of your suggestion, we have added a column named “data source” into Table 1 to specify and perfect the different data sources in this study. Revised Table 1 is as follows: 

Table 1. Variables definitions

Variable Variable name Variable code Measurement Data source

Explained

variables Short-term M&A performance CAR(-5，5) Cumulative abnormal returns of stock for the five days before and after the date of the M&A announcement CSMAR Database

RESSET Database

 Long-term M&A performance F1-F-1 The difference in comprehensive score of performance in the year before and after the date of the M&A announcement 

Explaining

variables Venture capital shareholding VC If there is venture capital shareholding among the top ten shareholders, the value is 1; otherwise, it is 0 CSMAR Database

WIND Database

Zero2IPO Database

 Venture capital shareholding ratio VCshares The sum of the shareholding ratio of venture capital institutions among the top ten shareholders 

 Venture capital syndicate VCsynd If the top ten stockholders include two or more venture capital firms, the value is 1; otherwise, it is 0 

Control

variables Relative size RS M&A transaction amount/total assets in the year before the M&A CSMAR Database

WIND Database

RESSET Database

 M&A type Type Take 1 for relevant M&A, 0 for others 

 Payment terms PT Take 1 for cash payment, 0 for others 

 Company size Size Natural logarithm of total assets in the year before the M&A 

 Company

growth Growth The growth rate of operating income in the year before the M&A 

 Liability level Lev Asset-liability ratio in the year before the M&A 

 Cashflow Cash FCF in the year before the M&A/total assets in the year before the M&A 

 Shareholding

concentration TOP5 The shareholding ratio of the top five shareholders in the year before the M&A 

 Earning

capacity Roa Return on total assets in the year before the M&A 

 Operation

capacity Tat Total asset turnover in the year before the M&A 

 Age of

company Age Natural logarithm of the number of years from the establishment of the company to the year of M&A 

 Industry dummy variable Ind The samples are divided into 14 industries, and 13 industry dummy variables are set. Guidelines on Industry Classification of Listed Companies in China [46]

 Year dummy variable Year The sample data is collected from three years, and two year dummy variables are set. 

6. I can not see the fig. of the research model. The authors should add it before the section of “3.2. Variables design”. 

Response: We would like to thank you again for your careful reading, helpful comments, and constructive suggestions, which have significantly improved the presentation of our manuscript. According to your comment, we have drawn a figure of the research model, which clearly demonstrates the framework and mechanism of our study, and added it before the section of “3.2 Variables design”. The figure is shown below. We have uploaded it as a separate file. 

Fig 1. Theoretical model

7. The authors should add the source into “Table 1. Variables definitions”.

Response: Thank you for your valuable and thoughtful comments. As a result of your suggestion, we added a column named “data source” into Table 1 to specify and perfect the different data sources in this study. Revised Table 1 is as follows: 

Table 1. Variables definitions

Variable Variable name Variable code Measurement Data source

Explained

variables Short-term M&A performance CAR(-5，5) Cumulative abnormal returns of stock for the five days before and after the date of the M&A announcement CSMAR Database

RESSET Database

 Long-term M&A performance F1-F-1 The difference in comprehensive score of performance in the year before and after the date of the M&A announcement 

Explaining

variables Venture capital shareholding VC If there is venture capital shareholding among the top ten shareholders, the value is 1; otherwise, it is 0 CSMAR Database

WIND Database

Zero2IPO Database

 Venture capital shareholding ratio VCshares The sum of the shareholding ratio of venture capital institutions among the top ten shareholders 

 Venture capital syndicate VCsynd If the top ten stockholders include two or more venture capital firms, the value is 1; otherwise, it is 0 

Control

variables Relative size RS M&A transaction amount/total assets in the year before the M&A CSMAR Database

WIND Database

RESSET Database

 M&A type Type Take 1 for relevant M&A, 0 for others 

 Payment terms PT Take 1 for cash payment, 0 for others 

 Company size Size Natural logarithm of total assets in the year before the M&A 

 Company

growth Growth The growth rate of operating income in the year before the M&A 

 Liability level Lev Asset-liability ratio in the year before the M&A 

 Cashflow Cash FCF in the year before the M&A/total assets in the year before the M&A 

 Shareholding

concentration TOP5 The shareholding ratio of the top five shareholders in the year before the M&A 

 Earning

capacity Roa Return on total assets in the year before the M&A 

 Operation

capacity Tat Total asset turnover in the year before the M&A 

 Age of

company Age Natural logarithm of the number of years from the establishment of the company to the year of M&A 

 Industry dummy variable Ind The samples are divided into 14 industries, and 13 industry dummy variables are set. Guidelines on Industry Classification of Listed Companies in China [46]

 Year dummy variable Year The sample data is collected from three years, and two year dummy variables are set. 

8. The authors should add the section of “discussion”. Meanwhile, the authors should add more to the “Discussion section”, meanwhile, comparing the result with previous ones. The authors should add the discussion section. At the present version, it is missing. It is illustrate he value and contribution of the manuscript.

Response: We are extremely grateful to the reviewer for pointing out this problem. As you suggested, we added the section of “discussion” to summarize existing research results and compare our findings with previous ones to emphasize the value and contribution of this manuscript. Specific expressions are as follows: 

6. Discussion

In the current relevant studies with a limited number, two incompatible research results are presented regarding the impact of venture capital shareholding on the M&A performance of GEM-listed companies: One conclusion suggests that both short-term and long-term M&A performance of GEM-listed companies with venture capital shareholding are superior; furthermore, an increase in venture capital shareholding markedly enhances the M&A performance of GEM-listed companies. The other research conclusion indicates that not only did venture capital shareholding fail to improve the long-term M&A performance of GEM-listed companies notably, but it also negatively affected their short-term M&A performance. This study provides a comprehensive analysis of the impact of venture capital institutions’ functions, such as certification, monitoring, and value-added services, on the crucial stages in the M&A process of GEM-listed companies and systematically clarifies the mechanism of venture capital shareholding on the M&A performance of GEM-listed companies. The empirical analysis based on this not only verifies the positive effect of venture capital shareholding on the short-term and long-term M&A performance of GEM-listed companies but also further explains the different effects of an increase in venture capital shareholding and a venture capital syndicate on the short-term and long-term M&A performance of GEM-listed companies. Compared with existing studies, this research more systematically demonstrates the mechanism by which venture capital shareholding affects the M&A performance of GEM-listed companies and analyzes the impact of venture capital shareholding on the short-term and long-term M&A performance of GEM-listed companies at a deeper level, addressing the shortcomings in current studies and providing a valuable reference for these companies to rationally understand and effectively utilize the positive effects of venture capital shareholders in M&A activities.

9. The authors should check the important and recent ones. 

Response: We would like to thank you for your careful reading, helpful comments, and constructive suggestions, which have significantly improved the presentation of our manuscript. According to the reviewer’s suggestion on the revision of references, we have replaced most of the literature in the manuscript with the latest ones. 

10. The authors should add limitation section. 

Response: We are very grateful to the reviewer for pointing out this concern. We have carefully considered your suggestions and added the “limitation” section at the end of the study to illustrate deficiencies of this research from three perspectives, which will be further taken into consideration in future research. Specific expressions of this section are as follows: 

7. Limitation

Although this paper explored the impact of venture capital shareholding on the M&A performance of GEM-listed companies from multiple perspectives, there are still several areas for improvement: (1) The capacity differences among venture capital institutions are objective. Limited by the studying materials, the specific impact of capacity difference among venture capital institutions on the M&A performance of GEM-listed companies is not further analyzed;(2) This study ana

---

## [Decision Letter · Decision Letter 1]

2 Aug 2024

Can venture capital shareholding improve M&A performance? An empirical study based on Chinese GEM-listed companies

PONE-D-24-04863R1

Dear Dr. Hu,

We’re pleased to inform you that your manuscript has been judged scientifically suitable for publication and will be formally accepted for publication once it meets all outstanding technical requirements.

Kind regards,

Juan E. Trinidad-Segovia, PhD

Section Editor

PLOS ONE

Additional Editor Comments (optional): **Comments from PLOS Editorial Office**: We note that one or more reviewers has recommended that you cite specific previously published works in an earlier round of revision. As always, we recommend that you please review and evaluate the requested works to determine whether they are relevant and should be cited. It is not a requirement to cite these works and you may choose to remove this before the manuscript proceeds to publication. We appreciate your attention to this request.

Reviewers' comments:

Reviewer's Responses to Questions

**Comments to the Author**

1. If the authors have adequately addressed your comments raised in a previous round of review and you feel that this manuscript is now acceptable for publication, you may indicate that here to bypass the “Comments to the Author” section, enter your conflict of interest statement in the “Confidential to Editor” section, and submit your "Accept" recommendation.

Reviewer #3: All comments have been addressed

2. Is the manuscript technically sound, and do the data support the conclusions?

Reviewer #3: Yes

3. Has the statistical analysis been performed appropriately and rigorously? 

Reviewer #3: Yes

4. Have the authors made all data underlying the findings in their manuscript fully available?

Reviewer #3: Yes

5. Is the manuscript presented in an intelligible fashion and written in standard English?

Reviewer #3: Yes

6. Review Comments to the Author

Reviewer #3: The author has satisfactorily responded to the reviewers' comments. The paper is remarkably clear, especially in the methodology and variables used. With the corrections and additions to the paper, the document is ready for publication,

7. PLOS authors have the option to publish the peer review history of their article (what does this mean?). If published, this will include your full peer review and any attached files.

Reviewer #3: No

---

## [Editor Report · Acceptance letter]

7 Aug 2024

PONE-D-24-04863R1 

PLOS ONE

Dear Dr. Hu, 

I'm pleased to inform you that your manuscript has been deemed suitable for publication in PLOS ONE. Congratulations! Your manuscript is now being handed over to our production team.

Kind regards, 

on behalf of

Dr. Juan E. Trinidad-Segovia 

Section Editor

PLOS ONE